# Incidence and Mortality of Renal Cell Carcinoma after Kidney Transplantation: A Meta-Analysis

**DOI:** 10.3390/jcm8040530

**Published:** 2019-04-17

**Authors:** Api Chewcharat, Charat Thongprayoon, Tarun Bathini, Narothama Reddy Aeddula, Boonphiphop Boonpheng, Wisit Kaewput, Kanramon Watthanasuntorn, Ploypin Lertjitbanjong, Konika Sharma, Aldo Torres-Ortiz, Napat Leeaphorn, Michael A. Mao, Nadeen J. Khoury, Wisit Cheungpasitporn

**Affiliations:** 1Department of Epidemiology, Harvard T.H. Chan School of Public Health, Boston, MA 02115, USA; api.che@hotmail.com; 2Department of Internal Medicine, Faculty of Medicine, Chulalongkorn University, Bangkok 10300, Thailand; 3Division of Nephrology and Hypertension, Mayo Clinic, Rochester, MN 55905, USA; 4Department of Internal Medicine, University of Arizona, Tucson, AZ 85721, USA; tarunjacobb@gmail.com; 5Division of Nephrology, Department of Medicine, Deaconess Health System, Evansville, IN 47747, USA; dr.anreddy@gmail.com; 6Department of Internal Medicine, East Tennessee State University, Johnson City, TN 37614, USA; boonpipop.b@gmail.com; 7Department of Military and Community Medicine, Phramongkutklao College of Medicine, Bangkok 10400, Thailand; wisitnephro@gmail.com; 8Department of Internal Medicine, Bassett Medical Center, Cooperstown, NY 13326, USA; kanramon@gmail.com (K.W.); ploypinlert@gmail.com (P.L.); drkonika@gmail.com (K.S.); 9Division of Nephrology, Department of Medicine, University of Mississippi Medical Center, Jackson, MS 39216, USA; Aldo_t86@hotmail.com (A.T.-O.); wcheungpasitporn@gmail.com (W.C.); 10Department of Nephrology, Department of Medicine, Saint Luke’s Health System, Kansas City, MO 64111, USA; napat.leeaphorn@gmail.com; 11Division of Nephrology and Hypertension, Mayo Clinic, Jacksonville, FL 32224, USA; mao.michael@mayo.edu; 12Department of Nephrology, Department of Medicine, Henry Ford Hospital, Detroit, MI 48202, USA; nadeenj.khoury@gmail.com

**Keywords:** malignancy, post-transplant malignancy, renal cell carcinoma, meta-analysis, kidney transplantation, transplantation, systematic reviews

## Abstract

Background: The incidence and mortality of renal cell carcinoma (RCC) after kidney transplantation (KTx) remain unclear. This study’s aims were (1) to investigate the pooled incidence/incidence trends, and (2) to assess the mortality/mortality trends in KTx patients with RCC. Methods: A literature search was conducted using the MEDLINE, EMBASE and Cochrane databases from inception through October 2018. Studies that reported the incidence or mortality of RCC among kidney transplant recipients were included. The pooled incidence and 95% CI were calculated using a random-effect model. The protocol for this meta-analysis is registered with PROSPERO; no. CRD42018108994. Results: A total of 22 observational studies with a total of 320,190 KTx patients were enrolled. Overall, the pooled estimated incidence of RCC after KTx was 0.7% (95% CI: 0.5–0.8%, *I*^2^ = 93%). While the pooled estimated incidence of de novo RCC in the native kidney was 0.7% (95% CI: 0.6–0.9%, *I*^2^ = 88%), the pooled estimated incidence of RCC in the allograft kidney was 0.2% (95% CI: 0.1–0.4%, *I*^2^ = 64%). The pooled estimated mortality rate in KTx recipients with RCC was 15.0% (95% CI: 7.4–28.1%, *I*^2^ = 80%) at a mean follow-up time of 42 months after RCC diagnosis. While meta-regression analysis showed a significant negative correlation between year of study and incidence of de novo RCC post-KTx (slopes = −0.05, *p* = 0.01), there were no significant correlations between the year of study and mortality of patients with RCC (*p* = 0.50). Egger’s regression asymmetry test was performed and showed no publication bias in all analyses. Conclusions: The overall estimated incidence of RCC after KTX was 0.7%. Although there has been a potential decrease in the incidence of RCC post-KTx, mortality in KTx patients with RCC has not decreased over time.

## 1. Introduction

Kidney transplantation (KTx) is the renal replacement therapy of choice for the majority of patients with end-stage renal disease (ESRD) and it significantly improves survival and quality of life [1,2]. The long-term mortality rate is 48% to 82% lower in KTx recipients when compared to ESRD patients on the transplant waitlist [2,3]. However, due to immunosuppression, KTx patients are at a two-fold increased risk of developing malignancy in comparison to the general population [4,5,6]. Malignancies are among the top three leading causes of death in KTx recipients, following infection and cardiovascular complications [6]. 

Studies have demonstrated a higher incidence of renal cell carcinoma (RCC) among ESRD patients (0.3%) than its reported incidence in the general population (approximately 0.005%) [7,8]. Thus, the Clinical Practice Guidelines Committee of the American Society of Transplantation (AST) [9] has recommended RCC screening for high-risk candidates, such as ESRD patients on dialysis for more than 3 years [10]. Despite screening for RCC among KTx candidates, de novo RCC has been reported among KTx patients in both native kidneys [11,12,13,14,15,16,17,18], and transplanted kidneys [17,19,20]. However, the incidence and incidence trends of RCC among KTx patients remain unclear [11,12,13,14,15,16,17,18,19,20,21,22,23,24,25,26,27,28,29,30,31,32,33,34,35,36,37,38,39,40,41,42]. 

Thus, we performed a systematic review to (1) investigate the pooled incidence/incidence trends, and (2) assess the mortality/mortality trends in KTx patients with RCC.

## 2. Methods

### 2.1. Search Strategy and Literature Review

The protocol for this systematic review is registered with PROSPERO (International Prospective Register of Systematic Reviews; no. CRD42018108994). A systematic literature search of MEDLINE (1946 to October 2018), EMBASE (1988 to October 2018), and the Cochrane Database of Systematic Reviews (database inception to October 2018) was performed to assess (1) the pooled incidence/incidence trends, and (2) to assess the mortality/mortality trends in KTx patients with RCC. The systematic literature review was conducted independently by two investigators (C.T. and W.C) using a search strategy that consolidated the terms “kidney cancer” OR “renal cell carcinoma” AND “kidney transplantation” OR “renal transplantation” which is provided in the online Appendix A. The database searches were limited to English language articles only. A manual search for conceivably related studies using references of the included articles was also performed. This study was conducted using the Preferred Reporting Items for Systematic Reviews and Meta-Analysis (PRISMA) statement [43].

### 2.2. Selection Criteria

Eligible studies had to be clinical trials or observational studies (cohort, case-control, or cross-sectional studies) that reported the incidence or mortality of RCC among adult KTx recipients (age >/= 18 years old). Retrieved articles were individually reviewed for eligibility by two investigators (A.C. and C.T.). Discrepancies were addressed and solved by mutual consensus. Inclusion was not limited by the size of study. 

### 2.3. Data Abstraction

A structured data collecting form was used to obtain the following information from each study: title, name of the first author, year of the study, publication year, country where the study was conducted, RCC definition, incidence of RCC, and mortality in KTx patients with RCC.

### 2.4. Statistical Analysis

Analyses were performed utilizing the Comprehensive Meta-Analysis 3.3 software (Biostat Inc, Englewood, NJ, USA). Adjusted point estimates from each study were consolidated by the generic inverse variance approach of DerSimonian and Laird, which designated the weight of each study based on its variance [44]. Given the possibility of between-study variance, we used a random-effect model rather than a fixed-effect model. Forest plots were constructed to visually evaluate the incidence and mortality of RCC among adult KTx recipients. Cochran’s Q test and *I*^2^ statistic were applied to determine the between-study heterogeneity. A value of *I*^2^ of 0–25% represents insignificant heterogeneity, 26–50% low heterogeneity, 51–75% moderate heterogeneity and 76–100% high heterogeneity [45]. The presence of publication bias was assessed using the Egger test [46]. Funnel plots were created to evaluate for the presence or absence of publication bias. 

## 3. Results

A total of 7815 potentially eligible articles were identified using our search strategy. After the exclusion of 7629 articles based on their title and abstract for clearly not fulfilling the inclusion criteria on the basis of type of article, patient population, study design, or outcome of interest, and 81 due to being duplicates, 105 articles were left for full-length review. Fifty-nine of them were excluded from the full-length review as they did not report the outcome of interest. Twenty-one articles were case reports and three articles were not in English. Thus, 22 cohort studies [11,12,13,14,15,16,17,18,19,20,23,28,29,30,31,32,33,34,35,36,38,39] with a total of 320,190 KTx patients were enrolled. The literature retrieval, review, and selection process are demonstrated in Figure 1. The characteristics of the included studies are presented in Table 1. 

### 3.1. Incidence of RCC after KTx

Eighteen studies provided data on the incidence of RCC after KTx [11,12,13,14,15,16,17,18,19,20,28,29,31,32,34,35]. Overall, the pooled estimated incidence of RCC after KTx was 0.7% (95% CI: 0.5–0.8%, *I*^2^ = 93%, Figure 2). While the pooled estimated incidence of de novo RCC in the native kidney was 0.7% (95% CI: 0.6–0.9%, *I*^2^ = 88%, Figure 3A), the pooled estimated incidence of RCC in the allograft kidney was 0.2% (95% CI: 0.1–0.4%, *I*^2^ = 64%, Figure 3B).

Meta-regression showed a significant negative correlation between year of study and incidence of de novo RCC post-KTx (slopes = −0.05, *p* = 0.01, Figure 4).

### 3.2. Mortality Rate in KTx Recipients with RCC

Eleven studies provided data the on mortality rate in KTx recipients with RCC [13,14,16,17,19,20,23,30,33,36,39]. Overall, the pooled estimated mortality rate in KTx recipients with RCC was 15.0% (95% CI: 7.4–28.1%, *I*^2^ = 80%, Figure 5) at a mean follow-up time of 42 months after RCC diagnosis. The data on the incidence and mortality of recurrent RCC among KTx recipients with a previous history of RCC prior to KTX were limited. A prior study demonstrated an incidence of recurrent RCC after KTX of 9.1% with an associated 5-year survival of 41.7% [23]. Sensitivity analysis, excluding the study of recurrent RCC among KTx recipients with a previous history of RCC prior to KTX (23), demonstrated a pooled estimated mortality rate of 11.5% in KTx recipients with RCC (95% CI: 6.4–19.8%, *I*^2^ = 67%).

Meta-regression showed no significant correlations between the year of study and mortality of patients with RCC (*p* = 0.50, Figure 6). When meta-regression was performed excluding the study of recurrent RCC among KTx recipients with a previous history of RCC prior to KTX [30], there were still no significant correlations between the year of study and mortality of patients with RCC (*p* = 0.56, Figure 7).

### 3.3. Evaluation for Publication Bias

Funnel plots (Appendix A) and Egger’s regression asymmetry tests were performed to evaluate publication bias in the analysis evaluating the incidence and mortality of KTx recipients with RCC. There was no significant publication bias, with *p*-values of 0.58 and 0.54, respectively. 

## 4. Discussion

In this systematic review, we found that RCC after KTx occurs with an incidence of 0.7%. RCC can occur in the native kidney with an incidence of 0.7% or in the allograft kidney with an incidence of 0.2%. Our findings also showed a statistically significant negative correlation between the incidence of RCC after KTx and study year, representing a potential decrease in the RCC incidence among KTx patients. However, mortality in KTx patients with RCC has not decreased over time. 

Post-KTx malignancy is a common cause of death [5,6,47,48,49,50,51] and RCC is the most common solid-organ malignancy in this population [52,53]. Due to the increased risk of RCC among ESRD patients [7,8], the Clinical Practice Guidelines Committee of the AST has suggested RCC screening in ESRD patients on dialysis for longer than 3 years [9,10]. In addition, it is suggested that most KTx candidates with a history of RCC should wait at least 2 years from successful cancer treatment to KTx (unless candidates have only small localized incidental tumors, which may not require any waiting period) [54,55]. Candidates with large, invasive or symptomatic RCC may require a longer waiting period of 5 years [54,55]. Despite RCC screening prior to KTx, the findings from our study suggest that RCC can still occur post-KTx at a higher incidence (0.7%) than its reported incidence among ESRD patients (0.3%) [8]. In addition, studies have demonstrated that KTx recipients have a relative increased risk of five- to ten-fold for RCC compared with an age-matched general population, and that the majority of these tumors arise in the setting of acquired kidney cystic disease (AKCD) which develops with chronic renal failure [5,8,35,56,57,58,59,60,61,62,63,64]. Although RCC occurrence is more frequent in the native kidneys of KTx recipients, RCC can also occur in the renal allograft (incidence of 0.2%) [17,19,20]. 

While the exact etiology of the increased risk of RCC in KTx remains unclear, it is likely linked to the immunosuppressed state [4]. Reported risk factors for post-KTx RCC include older age, male sex, African descent, excess body weight, smoking, hypertension, history of acquired cystic kidney disease (ACKD), previous RCC prior to KTx, and longer pre-transplant dialysis duration [3,6,18,23,29,31,34,35,65,66,67]. Studies have shown that causes of ESRD before KTx may also affect the incidence of post-KTx RCC [14,32,35]. While KTx recipients with ESRD due to glomerulonephritis, hypertensive nephrosclerosis, and vascular diseases have been shown to have a higher incidence of post-KTx RCC, recipients with ESRD due to diabetic nephropathy carry a lower risk of post-KTx RCC [14,32,35,68]. KTx recipients are usually under intensified medical surveillance and the higher incidence of RCC among KTx recipients compared to general populations and ESRD patients might be due to detection bias. On the other hand, the lack of consensual RCC screening among KTx recipients may also have underestimated the exact incidence among the KTx patient population. Currently, there are no universal recommendations for RCC screening among KTx patients [3,22,69,70,71,72]. While the European Renal Best Practice (ERBP) guidelines recommend native kidney ultrasound as RCC screening in kidney transplant recipients, and the European Association of Urology (EAU) recommends an annual ultrasound of native kidneys and allografts for anyone with ACKD, previous RCC, or von Hippel–Lindau disease [3,71,72], the Kidney Disease Improving Global Outcomes (KDIGO) and AST guidelines for post-KTx care currently do not suggest universal screening for RCC among KTx recipients [22,69,70]. Thus, there are various RCC screening approaches for KTx recipients at different transplant centers. Many cases of RCC have been discovered during investigations for post-transplant erythrocytosis, elevated serum creatinine, hematuria, urinary infection, or incidentally from imaging for other indications [33,73,74,75]. The majority of studies with available data on surveillance programs performed screening for RCC post-KTx annually by ultrasonography of native and allograft kidneys. Among KTx recipients with ACKD, acquired multicystic dysplasia, or a prior history of RCC required more frequent screenings, every 6 months [16,17,19,20,28,36,76]. Given that the risk is greatest in the first year post-KTx and the majority of RCCs occur in the first 5 years after KTx [15,29,31,65,77], previous reports suggest that KTx recipients should routinely undergo ultrasonography to screen RCC on the native kidney during the first 30 days post-KTx and every 5 years afterwards in the absence of renal cysts, or every 2 years in the presence of renal cysts [65,77,78,79]. Our study’s findings suggest the need for future studies to identify a cost-effective surveillance strategy for RCC among KTx recipients. This strategy would need to take into consideration both native and allograft kidneys, and differentiate KTx recipients with non-simple renal cysts [3,80]. 

Several limitations of our systematic review are worth mentioning. First, there are statistical heterogeneities in our meta-analysis. Potential sources for heterogeneities were the variations in the renal transplant recipient screening methods, patient characteristics, and differences in the immunosuppressive regimens used at various transplant centers, which may have affected the incidence of RCC and mortality rate in this population. Second, there is a lack of data from included studies on immunosuppressive regimens [81,82,83,84,85]. Mammalian target of rapamycin (mTOR) inhibitors have shown antineoplastic activities [86]. Although the effects of mTOR among KTx recipients have been shown mostly for non-melanoma skin cancer [87,88,89], future studies evaluating the effects of different immunosuppressive regimens on mortality in KTx patients with RCC are needed. Lastly, this is a meta-analysis of cohort studies and the data from population-based studies were limited. Thus, large population-based studies evaluating the incidence of RCC in KTx patients are required in the future.

In summary, the overall estimated incidence of RCC after KTX was 0.6%, with an associated high mortality rate in KTx recipients of 13.9%. Despite potential improvements in the post-KTx RCC incidence, the mortality in KTx patients with RCC has remained unchanged over time.

## Figures and Tables

**Figure 1 jcm-08-00530-f001:**
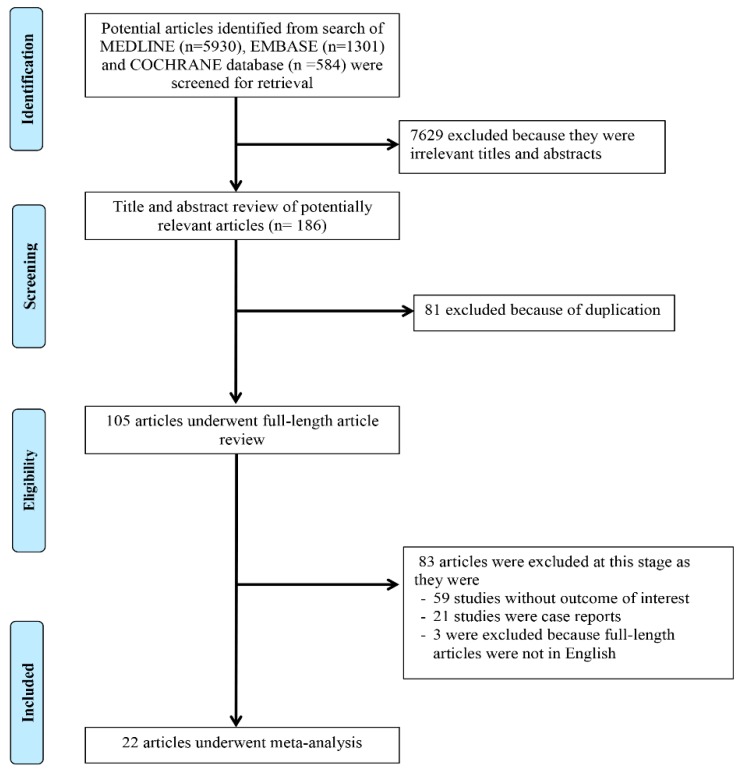
Outline of our search methodology.

**Figure 2 jcm-08-00530-f002:**
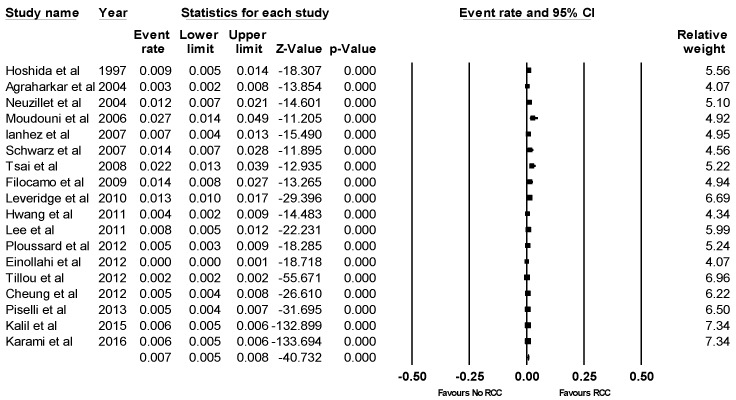
Forest plots of the included studies [11,12,13,14,15,16,17,18,19,20,28,29,31,32,34,35,38,39] assessing incidence rates of RCC after KTx. A diamond data marker represents the overall rate from each included study (square data marker) and 95% confidence interval.

**Figure 3 jcm-08-00530-f003:**
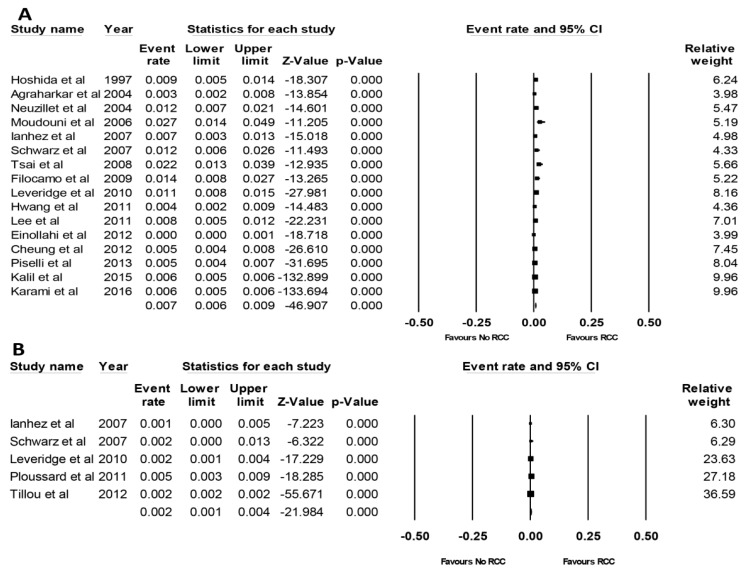
Forest plots of the included studies [11,12,13,14,15,16,17,18,28,29,31,32,34,35,38,39] assessing incidence rates of (**A**) de novo RCC in the native kidney and (**B**) RCC in the allograft kidney [17,19,20,38,39]. A diamond data marker represents the overall rate from each included study (square data marker) and 95% confidence interval.

**Figure 4 jcm-08-00530-f004:**
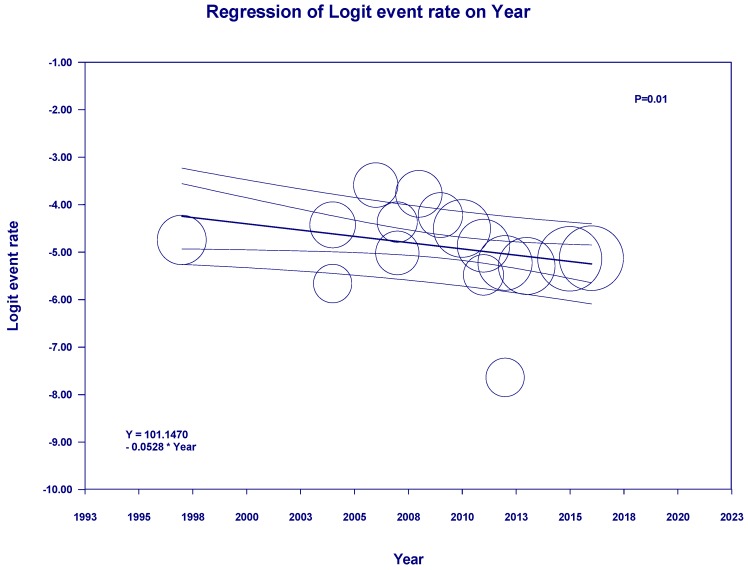
Meta-regression analyses showed a significant negative correlation between the year of study and incidence of de novo RCC post-KTx (slopes = −0.05, *p* = 0.01). The solid line represents the weighted regression line based on variance-weighted least squares. The inner and outer lines show the 95% confidence interval and prediction interval around the regression line. The circles indicate the log event rates in each study.

**Figure 5 jcm-08-00530-f005:**
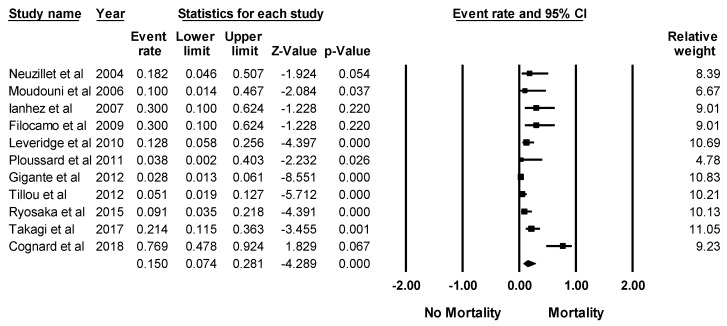
Forest plots of the included studies [13,14,16,17,19,20,23,30,33,36,39] assessing mortality rate in KTx recipients with RCC. A diamond data marker represents the overall rate from each included study (square data marker) and 95% confidence interval.

**Figure 6 jcm-08-00530-f006:**
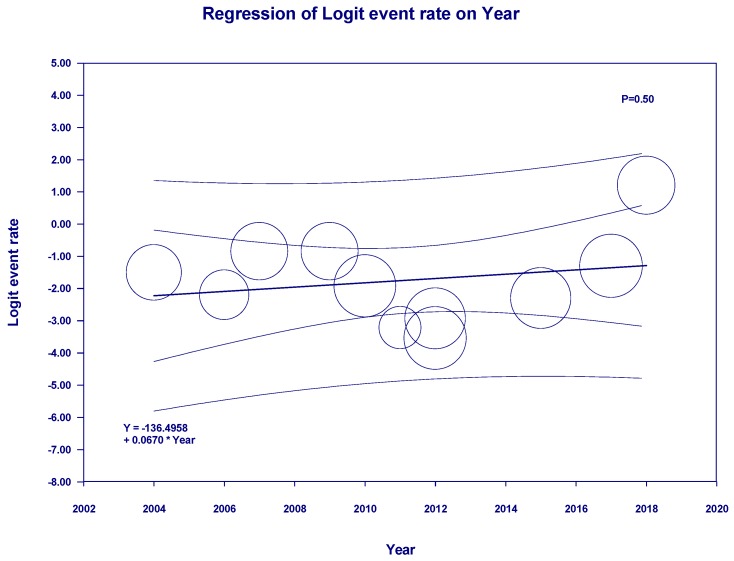
Meta-regression analyses showed no significant correlations between the year of study and mortality of patients with RCC (*p* = 0.50). The solid line represents the weighted regression line based on variance-weighted least-squares. The inner and outer lines show the 95% confidence interval and prediction interval around the regression line. The circles indicate the log event rates in each study.

**Figure 7 jcm-08-00530-f007:**
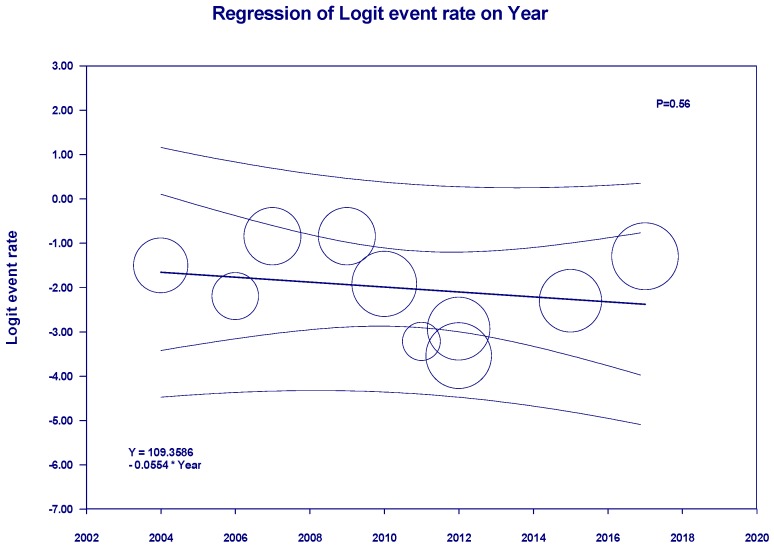
Meta-regression analyses, excluding the study of recurrent RCC among KTx recipients with a previous history of RCC prior to KTX, showed no significant correlations between the year of study and mortality of patients with RCC (*p* = 0.56). The solid line represents the weighted regression line based on variance-weighted least-squares. The inner and outer lines show the 95% confidence interval and prediction interval around the regression line. The circles indicate log event rates in each study.

**Table 1 jcm-08-00530-t001:** Main characteristics of studies included in the meta-analysis of incidence and mortality of renal cell carcinoma (RCC) after kidney transplantation (KTx) [11,12,13,14,15,16,17,18,19,20,23,28,29,30,31,32,33,34,35,36,38,39].

Study	Year	Type of Study	Number of Patients	Incidence of RCC	Follow-Up Time after Transplant	Time from Transplant to Cancer Diagnosis	Mortality of RCC	Quality Assessment
Hoshida et al. [11]	1997	Cohort	1744	15/1744	N/A	N/A	N/A	S4 C0 O3
Agraharkar et al. [12]	2004	Cohort	1739	6/1739	Mean 6.1 years	N/A	N/A	S4 C1 O3
Neuzillet et al. [13]	2004	Cohort	933	11/933	N/A	Mean 70.9 ± 49.4 (range 8–156) months	2/11 (1 died due to cancer)	S4 C0 O2
Moudouni et al. [14]	2006	Cohort	373	10/373	N/A	Mean 12.8 years Median 127 in patients treated with cyclosporine A and 114 months in patients not treated with cyclosporine A	1/10 (1 died due to cancer)	S4 C0 O3
Ianhez et al. [39]	2007	Cohort	1375	10/1375 9 in native kidney 1 in allograft kidney	N/A	N/A	3/10 (2 died due to myocardial infarction and one due to penile cancer)	S4 C0 O2
Schwarz et al. [38]	2007	Cohort	561	8/561 7 de novo in native kidney 1 de novo in allograft kidney	N/A	105.2 ± 62.39 months	N/A	S4C2O3
Tsai et al. [15]	2008	Cohort	3259	Touring group 15/215 kidney cancer Domestic group 4/321 kidney cancer	Touring group Mean 76.2 ± 48.1 months Domestic group Mean 81.5 ± 53.4 months	N/A	N/A	S4 C1 O3
Filocamo et al. [16]	2009	cohort	694	Native de novo 10/694	N/A	61.8 months (12–156 months)	3/10 (3 died due to cancer other than RCC)	S4 C1 O3
Leveridge et al. [17]	2010	cohort	3568	39/3568 native kidney 8/3568 allograft kidney	6.6 years	Native 10.6 years allograft 12.1 years	5 native died (not RCC cause), 1 allograft died due to cardiac cause	S4 C0 O3
Hwang et al. [18]	2011	Cohort	1695	7/1695	9.1 ± 6.9 years	Mean 11.8 ± 6.0 years	N/A	S4 C0 O3
Lee et al. [28]	2011	Cohort	2757	21/2757	N/A	Mean 119 (range 0–264) months	N/A	S4 C1 O2
Ploussard et al. [20]	2012	Cohort	2396	Allograft kidney 12/2396	N/A	Mean 13 (range 4–20) years	0/12	S4 C0 O3
Einollahi et al. [29]	2012	Cohort	12,525	6/12,525	N/A	Median 16 months	N/A	S4 C0 O3
Gigante et al. [30]	2012	Cohort	213	N/A	N/A	Mean 91 ± 82 months	6/213 due to RCC	S4 C0 O2
Tillou et al. [19]	2012	Cohort	41,806	Allograft kidney 79/41,806	N/A	Mean 131.7 (0.9–244) months	4/79	S4 C0 O3
Cheung et al. [31]	2012	Cohort	4895	26/4895	N/A	Median 4 (0.2–16.5) years	N/A	S4 C1 O3
Piselli et al. [32]	2013	Cohort	7217	31/7217	Median 5.2 years (2.9–7.8)	N/A	N/A	S4 C1 O3
Ryosaka et al. [33]	2015	Cohort	202	N/A	N/A	N/A	Solid-type renal cell carcinoma 2/17 Cystic-type renal cell carcinoma 2/27	S4 C0 O3
Kalil et al. [34]	2015	Cohort	115,845	Primary kidney transplant 514/109,224 Retransplant 43/6621	Mean 1st–4.6 years 2nd–3.7 years 3rd–2.9 years 4th–3.4 years	N/A	N/A	S4 C2 O3
Karami et al. [35]	2016	Cohort	116,208	683/116,208	Median 4.2 years (range 0.003–23.1)	N/A	N/A	S4 C0 O2
Takagi et al. [36]	2017	Cohort	42	N/A	N/A	Mean 86 ± 69 months	9/42 (5 died due to cancer)	S4 C0 O3
Cognard et al. [23]	2018	Cohort	143 with history of pre-transplant kidney cancer	13/143	Mean 5.6 ± 3.2 years	Mean 3 ± 2.3 years (range 45 days–7 years)	10/13 (9 died due to cancer)	S4 C0 O3

KTx, kidney transplantation; N/A, not available; RCC, renal cell carcinoma; S, C, O, selection, comparability, and outcome.

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
