# Peer review of "Incidence and Mortality of Renal Cell Carcinoma after Kidney Transplantation: A Meta-Analysis"

_jcm, 2019, doi:10.3390/jcm8040530_

Reviewer 1 Report

This study presents a systematic review to investigate the pooled incidence/incidence trends and assess the mortality/mortality trends in kidney transplantation (KT) patients with renal cell carcinoma (RCC). The authors showed that although there have been potential improvements in incidence of RCC post KT, mortality in KT patients with RCC has not decreased over time. Below are some minor comments for the authors to address. The paper looks great! Thank you.

Minor:

I would like to ask the authors to add I^2 values in the Result section in Abstract to show the heterogeneity between studies.

In Statistical Analysis section, please explain the visualization methods, Forest and Funnel plots.

Line 150, please add the correlation plot similar to Figures 4 and 6 for the meta regression to assess the correlation between the year of study and mortality of patients with RCC. 

Typo:

Line 73, please add “1)” before to assess the pooled …

Line 97, word pos-25sibility?

Line 99, please add 25% after 0%.

Author Response

Response to Reviewer#1

This study presents a systematic review to investigate the pooled incidence/incidence trends and assess the mortality/mortality trends in kidney transplantation (KT) patients with renal cell carcinoma (RCC). The authors showed that although there have been potential improvements in incidence of RCC post KT, mortality in KT patients with RCC has not decreased over time. Below are some minor comments for the authors to address. The paper looks great! Thank you.

Response: We thank you for reviewing our manuscript and for your critical evaluation.

Comment #1

 I would like to ask the authors to add I2 values in the Result section in Abstract to show the heterogeneity between studies.

Response:  We appreciate the reviewer’s input. We agree with the reviewer and added I2 values in the Result section in Abstract as the reviewer’s suggestion.

Comment #2

 In Statistical Analysis section, please explain the visualization methods, Forest and Funnel plots.

Response:  We agree with the reviewer’s important comment. Thus, we have added the description of forest and funnel plots and the visualization methods in the statistical analysis section as the reviewer’s suggestion.

Comment #3

Line 150, please add the correlation plot similar to Figures 4 and 6 for the meta-regression to assess the correlation between the year of study and mortality of patients with RCC.

Response:  We agree with the reviewer’s suggestion. We have added the correlation plot for the meta-regression to assess the correlation between the year of study and mortality of patients with RCC (as new Figure 6), as the reviewer’s suggestion.

Comment #4

Line 73, please add “1)” before to assess the pooled …

Response:  We appreciated the reviewer thorough review. We have added “1)” as the reviewer’s suggestion.

Comment #5

Line 97, word pos-25sibility?

Response:  We apologize for misspelled word. We have corrected to “possibility”

Comment #6

Line 99, please add 25% after 0%.

Response:  We appreciated the reviewer thorough review. We have made correction and changed to “I2 of 0%-25%” as the reviewer’s suggestion.

We appreciated the reviewer’s comment and have added these discussions in our manuscript as the reviewer’s suggestion.

Reviewer 2 Report

A well researched and presented article. Line 170 should read “representing a potential decrease in incidence” rather than “improvement” which I find confusing.

 I have a few comments that should be expanded upon in the discussion.

1.       What is the contribution of pre-transplant immunosuppression exposure to the incidence of post-transplant RCC? Is this the explanation for the high incidence of RCC in native kidneys compared to the general population?  Is the fall in the incidence of RRC post-transplant related to decreased use of cyclophosphamide and azathioprine for primary GNs for example? Is the incidence of RCC post-transplant the same for pts with diabetes as the cause of renal failure compared to primary GN? Is the incidence of RCC higher in pts with ADPKD as the cause of renal failure compared to other disease?

2.       What percentage of the studies screened pts for RCC post-transplant ? Is this the explanation for the higher incidence of RCC post-transplant compared to a non-screened age-matched population?  What is the incidence of RCC in the general population if screening is applied to high risk patients?

3.       Is mortality of RCC the same in studies that screened for RCC post transplantation versus studies that did not screen post transplantation?

4.       Is there any information about the success or not of screening post transplantation for other solid organ malignancies? My understanding that screening for bowel cancer for example post transplantation was not shown to improve survival in low risk population.

5.       What would a cost effective surveillance program look like based on the studies you have examined? Might be worth making a comment about this as well. Annual versus second yearly etc.

Author Response

Response to Reviewer #2

A well researched and presented article.

Response: We thank you for reviewing our manuscript and for your critical evaluation.

Comment #1

 Line 170 should read “representing a potential decrease in incidence” rather than “improvement” which I find confusing.

Response:  We agree with the reviewer’s important comment. We have changed to “representing a potential decrease in the RCC incidence among KTx patients” as the reviewer’s suggestion.

Comment #2

I have a few comments that should be expanded upon in the discussion.

What is the contribution of pre-transplant immunosuppression exposure to the incidence of post-transplant RCC? Is this the explanation for the high incidence of RCC in native kidneys compared to the general population?  Is the fall in the incidence of RRC post-transplant related to decreased use of cyclophosphamide and azathioprine for primary GNs for example? Is the incidence of RCC post-transplant the same for pts with diabetes as the cause of renal failure compared to primary GN? Is the incidence of RCC higher in pts with ADPKD as the cause of renal failure compared to other disease?

Response:  The reviewer raised very important point. Among kidney transplant patients with RCC, Studies have shown that causes of ESRD before KTx may also affect the incidence of post KTx RCC. While KTx recipients with ESRD due to glomerulonephritis, hypertensive nephrosclerosis, and vascular diseases have been shown to have higher incidence of post KTx RCC, recipients with ESRD due to diabetic nephropathy carry a lower risk of post KTx RCC. Data on RCC risk among KTx recipients with ADPKD are conflicting. While Piselli et al demonstrated that ADPKD is the second cause of ESRD leading to transplant among KTx patients with RCC, Karami et al found recipients with polycystic kidney disease had reduced risk of clear cell RCC. The data on the impacts of decreased use of cyclophosphamide and azathioprine for primary GNs on the incidence of RCC among KTX were limited. However, it is a possibility as the reviewer raised. We reviewed all included studies again for reported risk factors for RCC post KTx, which include older age, male sex, African descent, excess body weight, smoking, hypertension, history of acquired cystic kidney disease, previous RCC prior to KTx, and longer pre-transplant dialysis duration

We appreciated the reviewer’s excellent point and we have included this important point in our manuscript as the reviewer’s suggestion.

Comment #3

 What percentage of the studies screened pts for RCC post-transplant? Is this the explanation for the higher incidence of RCC post-transplant compared to a non-screened age-matched population?  What is the incidence of RCC in the general population if screening is applied to high risk patients?

Is mortality of RCC the same in studies that screened for RCC post transplantation versus studies that did not screen post transplantation?

Response:  We appreciate the reviewer’s input. We agree this is an important point. Thus, we reviewed all included studies again. There are a limited number of studies that provided the information on RCC screening (Filocamo et al, Ianhez et al, Schwarz et al, Leveridge et al, Lee et al, Ploussard et al, and Tillou et al). The other studies did not provide information on RCC screening among KTx recipients. We additionally performed meta-analysis based on the subgroup of studies with vs without data on RCC screening. Overall, the pooled incidence of post KTx RCC was 0.7% (95%CI 0.3%-1.6%) in studies with RCC screening, and 0.6% (95%CI 0.5%-0.8%) in studies without data on RCC screening.

Because the screening protocols for RCC post transplantation were not clear and varied so much due to the lack of recommendations, we could not perform any subgroup analysis based on screening RCC to see the mortality of RCC in each subgroup.

Given limited data on screening for RCC in a number of included studies, we additionally included this point as additional limitation of our study.

“KTx recipients are usually under intensified medical surveillance. Thus, the higher incidence of RCC among KTx recipients than general populations, and ESRD patients might be due to detection bias. On the other hand, the lack of consensual RCC screening among KTx recipients may have also underestimated the exact incidence among KTx patent population.”

Comment #4

Is there any information about the success or not of screening post transplantation for other solid organ malignancies? My understanding that screening for bowel cancer for example post transplantation was not shown to improve survival in low risk population.

Response: We appreciated the reviewer’s important comment. We agree that more studies are needed to evaluate if post-transplant surveillance for cancers would improve patient survival. Although the current evidence was limited, the Clinical Practice Guidelines Committee of the American Society of Transplantation has published recommended guidelines for cancer screening in renal transplant patients suggesting that screening for breast cancer among women 50 to 69 years and among ≥70 years of age whose life expectancy is more than 8 years would yield a better outcome and skin cancer which needed to self-examine monthly.

Comment #5

What would a cost effective surveillance program look like based on the studies you have examined? Might be worth making a comment about this as well. Annual versus second yearly etc.

Response: Thank you for raising this interesting point and it would impact a lot for kidney transplant patients. Although the cost effectiveness study on this topic were lacking, we reviewed all included studies again and gathered data on proposed surveillance program/screening for post KTx RCC. 

Majority of studies with available data on surveillance program performed screening for RCC post KTx annually by ultrasonography of native and allograft kidneys.

Among KTx recipients with acquired cystic kidney disease, acquired multicystic dysplasia, prior history of RCC required more frequent every six months (Filocamo et al, Leveridge et al, Lee et al, Ploussard et al, Tillou et al,  Takagi et al).

Given the risk is greatest in the first year post KTx and majority of RCCs occurs in the first 5 years after KTx (Tsai et al 2011, Einollahi 2012, and Cheung et al. 2012), previous reports suggest that KTx recipients should routinely undergo ultrasonography to screen RCC on the native kidney during the first 30 days post KTx and every five years afterwards in the absence of renal cysts, or every two years in the presence of renal cysts (Goh et al 2011 and Klatte et al 2010). Future studies of cost and benefits of screening for post KTx RCC are required.

We appreciated the reviewer’s comment and have added these discussions in our manuscript as the reviewer’s suggestion.
